# VARIATIONAL AUTOENCODERS TRAINED WITH Q-DEFORMED LOWER BOUNDS

## ABSTRACT

Variational autoencoders (VAEs) have been successful at learning a low-dimensional manifold from high-dimensional data with complex dependencies. At their core, they consist of a powerful Bayesian probabilistic inference model, to capture the salient features of the data. In training, they exploit the power of variational inference, by optimizing a lower bound on the model evidence. The latent representation and the performance of VAEs are heavily influenced by the type of bound used as a cost function. Significant research work has been carried out into the development of tighter bounds than the original ELBO, to more accurately approximate the true log-likelihood. By leveraging the q-deformed logarithm in the traditional lower bounds, ELBO and IWAE, and the upper bound CUBO, we bring contributions to this direction of research. In this proof-of-concept study, we explore different ways of creating these q-deformed bounds that are tighter than the classical ones and we show improvements in the performance of such VAEs on the binarized MNIST dataset.

## 1 INTRODUCTION

Variational autoencoders (VAEs) ((Rezende et al., 2014), (Kingma & Welling, 2014)) are powerful Bayesian probabilistic models, which combine the advantages of neural networks with those of Bayesian inference. They consist of an encoder created with a neural network architecture, which maps the high-dimensional input data, $\boldsymbol{x}$, to a low-dimensional latent representation, $\boldsymbol{z}$, through the posterior probability distribution, $p(\boldsymbol{z}|\boldsymbol{x})$. Then, samples from this latent distribution are decoded back to a high-dimensional signal, through another neural network architecture and the probability distribution $p(\boldsymbol{x}|\boldsymbol{z})$. Integration performed with these probability distributions from the Bayesian framework of VAEs is intractable. As a solution, variational inference is employed to perform learning in these models, whereby a tractable bound on the model evidence is optimized instead of the intractable model evidence itself (Jordan et al., 1999). By design, the output model is set as $p(\boldsymbol{x}|\boldsymbol{z})$, usually a Bernoulli or a Gaussian probability distribution, depending on whether the target is discrete or continuous, and the prior distribution of the latent space as $p(\boldsymbol{z})$. However, the true posterior distribution, $p(\boldsymbol{z}|\boldsymbol{x})$, remains unknown and is intractable. To solve this issue, an approximate posterior distribution, $q(\boldsymbol{z}|\boldsymbol{x})$, is learnt by means of a lower bound on the model evidence, termed the ELBO. For one data point, $\boldsymbol{x}^{(i)}$, writing out the Kullback-Leibler divergence between the true and approximate posterior distributions and using its positivity property yields this bound:

$$\log p(\boldsymbol{x}^{(i)}) \geq \text{ELBO} = \mathbb{E}_{q(\boldsymbol{z}|\boldsymbol{x}^{(i)})}\left[\log \frac{p(\boldsymbol{x}^{(i)}|\boldsymbol{z}) \cdot p(\boldsymbol{z})}{q(\boldsymbol{z}|\boldsymbol{x}^{(i)})}\right]$$
$$= \mathbb{E}_{q(\boldsymbol{z}|\boldsymbol{x}^{(i)})}\left[\log p(\boldsymbol{x}^{(i)}|\boldsymbol{z})\right] - D_{KL}(q(\boldsymbol{z}|\boldsymbol{x}^{(i)})||p(\boldsymbol{z})). \quad (1)$$

The lower bound on the model evidence, the ELBO, now becomes the cost function used during the training phase of the VAEs. Over time, the first term shows how the reconstruction loss changes and the second term how far the approximate posterior is to the prior distribution. The result of inference and the performance of VAEs on reconstructing and generating images heavily depend on the type of bound employed in training. A significant body of work has been carried out to replace the ELBO with tighter bounds on the model evidence. On the one hand, starting from an unbiased estimator of

the true log-likelihood, the authors of (Burda et al., 2016) derive an importance sampling estimate of the model evidence, the IWAE. This represents one of the tightest bounds of VAEs and has only recently been improved on in (Rainforth et al., 2018), (Tao et al., 2018). Increasing the number of importance samples in the IWAE objective, decreases the signal-to-noise-ratio of the gradients, which makes the learning more difficult, as the gradients suffer from a larger level of noise (Rainforth et al., 2018). Several strategies are able to correct this issue. In the first algorithm, MIWAE, the outer expectation of the IWAE objective is approximated with more than one sample, as is the case in the IWAE. The second algorithm, CIWAE, represents a convex combination of the ELBO and the IWAE bounds and the third algorithm, PIWAE, separately trains the encoder and the decoder networks with different IWAE objectives.

On the other hand, leveraging different divergences between the true and the approximate posterior distributions has lead to diverse bounds on the model evidence. Starting from the Rényi $\alpha$-divergence (Rényi, 1961) between such distributions, a family of lower and upper bounds are obtained, parameterized by $\alpha$ (Li & Turner, 2016). However, these lower bounds become competitive with the IWAE, only in the limit $\alpha \to -\infty$. In addition, the upper bounds suffer from approximation errors and bias and the means to select the best value of the hyperparameter $\alpha$ is unknown. Through an importance sampling scheme similar to the one found in the IWAE, these Rényi $\alpha$ bounds are tightened in (Webb & Teh, 2016). If the Rényi $\alpha$-divergence is replaced with the $\chi^2$ divergence, the bound on the model evidence becomes the upper bound CUBO (Dieng et al., 2017). The Rényi $\alpha$-family of bounds and others lose their interpretability as a reconstruction loss and a Kullback-Leibler divergence term that measures how close the approximate posterior is to the prior distribution. They remain just a cost function optimized during training.

With different compositions of convex and concave functions, the approaches described above are unified in the K-sample generalized evidence lower bound, GLBO (Tao et al., 2018). This study generalizes the concept of maximizing the logarithm of the model evidence to maximizing the $\phi$-evidence score, where $\phi(u)$ is a concave function that replaces the logarithm. It allows for great flexibility in the choice of training objectives in VAEs. One particular setting provides a lower bound, the CLBO, which surpasses the IWAE objective.

## 1.1 OUR CONTRIBUTIONS

The aim of this work is to leverage the theory of q-deformed functions introduced in (Tsallis, 1988), (Tsallis, 1994), (Tsallis, 1998), to derive tighter lower bounds on the model evidence in VAEs. To this end, our contributions are three-fold: firstly, we derive two novel lower bounds, by replacing the logarithm function in the classical ELBO, (Rezende et al., 2014), (Kingma & Welling, 2014), and IWAE bounds, (Burda et al., 2016), (Mnih & Rezende, 2016), respectively, with the q-deformed logarithm function. Values of $q < 1.0$ yield upper bounds of varying tightness on the classical logarithm function, as illustrated in Figure 1.

Secondly, we combine the information given by the upper bound CUBO, (Dieng et al., 2017), with the information given by the ELBO and the IWAE, respectively, to obtain a lower bound that is placed between the two. By the means of their construction, we hypothesize these q-deformed bounds to be closer to the true log-likelihood. We are able to confirm it in our experiments. We term our novel lower bounds the qELBO and the qIWAE.

Thirdly, the tightness of the gap between the classical logarithm function and the q-deformed one depends on the value of q, as seen in Figure 1. Thus, q becomes a hyperparameter of our algorithm. Since q is a number, we can optimize it efficiently and accurately, using standard optimization algorithms. By solving for the best q for each data batch, we make q a data-driven hyperparameter, tuned in an adaptive way during training.

## 2 METHODS

With the q-entropy, introduced in Tsallis (1988), the author developed the field of *nonextensive statistical mechanics*, as a generalization of traditional statistical mechanics, centered around the Boltzmann-Gibbs distribution. The $S_q$ entropy provides a generalization of this distribution, which can more accurately explain the phenomena of anomalous physical systems, characterized by rare events. In the following definitions, the original quantities can be recovered in the limit $q \to 1$. If

$k > 0$ is a constant, $W \in \mathbb{N}$ is the total number of possible states of a system and $p_i$ the corresponding probabilities, $\forall i = 1 : W$, then:

$$S_q = k \cdot \frac{1 - \sum_{i=1}^{W} p_i^q}{q - 1}, q \in \mathbb{R}. \tag{2}$$

The generalized logarithmic function, termed the q-logarithm, is introduced in Tsallis (1994) as:

$$\log_q(x) = \frac{x^{1-q} - 1}{1 - q}, \forall x, q \in \mathbb{R}. \tag{3}$$

The Kullback-Leibler divergence is generalized in Tsallis (1998) to the form

$$\mathrm{KL}_q(p||p_0) = -\int_x p(x) \cdot \frac{\left[\frac{p_0(x)}{p(x)}\right]^{1-q} - 1}{1 - q} \mathrm{dx} = \int_x p(x) \cdot \frac{\left[\frac{p(x)}{p_0(x)}\right]^{q-1} - 1}{q - 1} \mathrm{dx}. \tag{4}$$

In order to derive our q-deformed bounds, we replace the logarithm function from the ELBO and

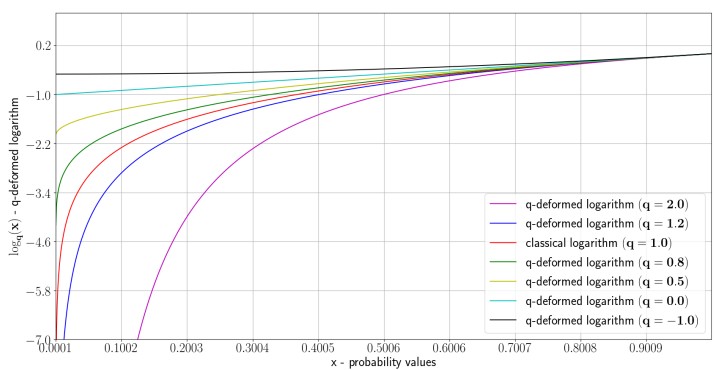

Figure 1: The $q$-deformed logarithm plotted for different values of the parameter $q$. Lower and upper bounds on the classical logarithm ($q = 1.0$) can be obtained depending on q, as well as the tightness of this gap.

IWAE bounds, with its q-deformed version. By appropriately optimizing the hyperparameter q, we will obtain an upper bound on the ELBO and IWAE, respectively:

$$\mathrm{ELBO} = \mathbb{E}_{q(\boldsymbol{z}|\boldsymbol{x})} \left[ \log \frac{p(\boldsymbol{x}|\boldsymbol{z}) \cdot p(\boldsymbol{z})}{q(\boldsymbol{z}|\boldsymbol{x})} \right],$$

$$\mathrm{qELBO} = \mathbb{E}_{q(\boldsymbol{z}|\boldsymbol{x})} \left[ \log_q \frac{p(\boldsymbol{x}|\boldsymbol{z}) \cdot p(\boldsymbol{z})}{q(\boldsymbol{z}|\boldsymbol{x})} \right] = \mathbb{E}_{q(\boldsymbol{z}|\boldsymbol{x})} \left\{ \frac{\left[\frac{p(\boldsymbol{x}|\boldsymbol{z}) \cdot p(\boldsymbol{z})}{q(\boldsymbol{z}|\boldsymbol{x})}\right]^{1-q} - 1}{1 - q} \right\}, \tag{5}$$

$$\mathrm{IWAE} = \mathbb{E}_{\boldsymbol{z}_1, \ldots, \boldsymbol{z}_K \sim q(\boldsymbol{z}|\boldsymbol{x})} \left[ \log \frac{1}{K} \sum_{i=1}^{K} \frac{p(\boldsymbol{x}|\boldsymbol{z}_i) \cdot p(\boldsymbol{z}_i)}{q(\boldsymbol{z}_i|\boldsymbol{x})} \right]$$

$$\mathrm{qIWAE} = \mathbb{E}_{\boldsymbol{z}_1, \ldots, \boldsymbol{z}_K \sim q(\boldsymbol{z}|\boldsymbol{x})} \left[ \log_q \frac{1}{K} \sum_{i=1}^{K} \frac{p(\boldsymbol{x}|\boldsymbol{z}_i) \cdot p(\boldsymbol{z}_i)}{q(\boldsymbol{z}_i|\boldsymbol{x})} \right]$$

$$= \mathbb{E}_{\boldsymbol{z}_1, \ldots, \boldsymbol{z}_K \sim q(\boldsymbol{z}|\boldsymbol{x})} \left\{ \frac{\left[\frac{1}{K} \sum_{i=1}^{K} \frac{p(\boldsymbol{x}|\boldsymbol{z}_i) \cdot p(\boldsymbol{z}_i)}{q(\boldsymbol{z}_i|\boldsymbol{x})}\right]^{1-q} - 1}{1 - q} \right\}. \tag{6}$$

**Optimization algorithm for q**. We train a variational autoencoder with our novel qELBO and qIWAE bounds. The training procedure and the optimization method for q are identical for both types of q-deformed bounds. We will describe them in the case of the qELBO.

We start the training procedure with an initial value of $q = 1.0 - 10^{-6}$. For one batch of images, we compute the qELBO lower bound and the CUBO upper bound (Dieng et al., 2017), averaged over the batch. In order to obtain a tighter lower bound, qELBO$^*$, we set a desired value of the cost function at

$$\text{qELBO}^* = \text{qELBO} + \tau \cdot (\text{CUBO} - \text{qELBO}), \text{ where, in our experiments, } \tau \in \{0.5, 0.75\}.$$

By means of the L-BFGS-B optimization method, we find the optimal value $q^*$, such that

$$\text{qELBO}^* = \mathbb{E}_{q(\boldsymbol{z}|\boldsymbol{x})} \left\{ \frac{\left[ \frac{p(\boldsymbol{x}|\boldsymbol{z}) \cdot p(\boldsymbol{z})}{q(\boldsymbol{z}|\boldsymbol{x})} \right]^{1-q^*} - 1}{1 - q^*} \right\}. \tag{7}$$

For this task, we employ the scipy optimization package in python. We apply the gradient descent step on our new, improved, cost function, qELBO$^*$, computed with this optimal value, $q*$. We save this value of $q$ for the next batch of images and we repeat the optimization steps described above, for all training batches.

## 3 EXPERIMENTAL RESULTS

### 3.1 NEURAL NETWORK ARCHITECTURE

For the experiments conducted on the MNIST dataset (LeCun et al., 1998), we use the one-stochastic layer architecture employed in (Burda et al., 2016) and in (Li & Turner, 2016). The encoder and the decoder are composed of two deterministic layers, each with 200 nodes, and of a stochastic layer with 50 nodes. The dimension of the latent space is equal to 50 and the activation functions are the softplus function. The approximate posterior is modeled as a Gaussian distribution, with a diagonal covariance matrix. The output model is a Bernoulli distribution for each pixel. We use the binarized MNIST dataset provided by tensorflow, with 55000 training images and 10000 test images. The learning rate is fixed at 0.005 and there is no updating schedule. To implement and test our new algorithms, we modify publicly available code [1] (Li & Turner, 2016).

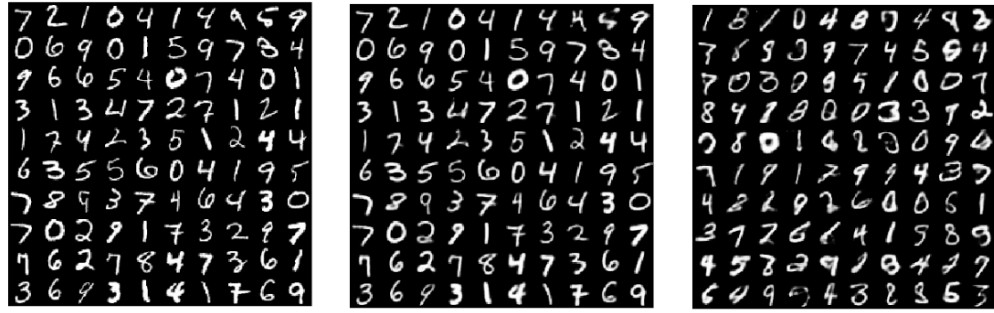

Figure 2: Method: VAE with K=50 samples. From left to right: original binary MNIST test images, reconstructed and randomly generated ones.

### 3.2 RESULTS

On the benchmark binary MNIST dataset (LeCun et al., 1998), we compare our newly derived q-deformed bounds with the ELBO and the IWAE and we show several improvements that we obtained. On the test set, we report the bounds computed with K number of samples and the true log-likelihood estimated with 5000 importance samples, $\log \hat{p}_x$. The expectations involved in all

---

[1] https://github.com/YingzhenLi/vae_renyi_divergence

Table 1: Log-likelihood results on the binarized MNIST dataset, after 3000 epochs of training: the bound estimated with K number of samples used in training and the true log-likelihood estimated with 5000 importance samples ($\log \hat{p}_x$). The results are reported on the test set.

| METHOD | Bound with K samples | $\log \hat{p}_x$(5000 samples) | K | No. of epochs |
|---|---|---|---|---|
| VAE | -96.01 | -91.32 | 50 | 3000 |
| qVAE ($\tau$ =0.5) | **-91.02** | **-91.10** | 50 | 3000 |
| qVAE ($\tau$ =0.75) | -91.4 | -91.51 | 50 | 3000 |
| | | | | |
| IWAE | -91.53 | -89.72 | 50 | 3000 |
| qIWAE ($\tau$ =0.5) | -89.12 | **-89.68** | 50 | 3000 |
| qIWAE ($\tau$ =0.75) | **-88.90** | -89.77 | 50 | 3000 |
| | | | | |
| VAE | -96.09 | -91.34 | 5 | 3000 |
| qVAE($\tau$ =0.5) | -93.22 | -91.23 | 5 | 3000 |
| qVAE($\tau$ =0.75) | **-92.91** | **-91.07** | 5 | 3000 |
| | | | | |
| IWAE | -93.56 | -90.30 | 5 | 3000 |
| qIWAE ($\tau$ =0.5) | -91.6 | -90.28 | 5 | 3000 |
| qIWAE ($\tau$ =0.75) | **-91.22** | **-90.21** | 5 | 3000 |

of the bounds are estimated with Monte Carlo sampling. For the ELBO and the qELBO bounds, the expectation is approximated with K number of samples. The expectation in the standard IWAE is approximated with one sample. Thus, we will compute the expectation in the qIWAE with one sample, as well. Here, K refers to the number of importance samples used in the computation of the bound. In addition, we illustrate the performance of our algorithms on reconstructed binary MNIST test images and on randomly generated ones. After 3000 epochs of training, the qIWAE($\tau$ =0.5)

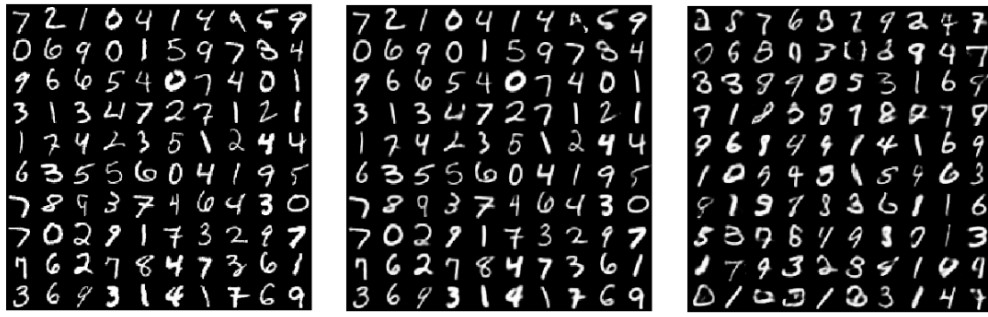

Figure 3: Method: qVAE($\tau = 0.5$) with K=50 samples. From left to right: original binary MNIST test images, reconstructed and randomly generated ones.

algorithm, with the bound estimated with K=50 samples, gives the best result on the importance sampling estimate of the true log-likelihood, very close to the one given by the standard IWAE. Moreover, the q-deformed bound is much closer to the estimated true value, than is the IWAE bound. We observe this behaviour for all the q-deformed bounds. This implies that, during training, optimizing the q-deformed bounds provides a cost function that is a more accurate approximation of the model evidence. Although the q-deformed ELBO does not outperform the standard IWAE, we can see significant improvements over the traditional ELBO, in all the test cases. A large decrease in the value of the bound is present for all the qELBO variants, more pronounced in the large sample regime.

## 4    CONCLUSION AND FUTURE WORK

We addressed the challenging task of deriving tighter bounds on the model evidence of VAEs. Significant research effort has gone in this direction, with several major contributions having been developed so far, which we reviewed in the introduction. We leveraged the q-deformed logarithm function, to explore other ways of tightening the lower bounds. As well as improvements in the estimated true log-likelihood, we found that the q-deformed bounds are much closer to the estimated true log-likelihood, than the classical bounds are. Thus, training with our novel bounds as the cost function may increase the learning ability of VAEs. Through the preliminary experiments we have conducted so far, we have achieved our goal. They show that our approach has merit and that this direction of research is worth pursuing in more depth, to produce more accurate bounds and to study their impact on the performance of VAEs.

As future work, similarly to (Rainforth et al., 2018), we plan to investigate how the tightening the ELBO and the IWAE influences the learning process and affects the gradients and the structure of the latent space, compared with the classical case. In addition, we plan to explore different optimization strategies for q and to study its role in achieving tighter bounds. We will also apply our q-deformed bounds, to investigate the disentanglement problem in VAEs, see for example (Higgins et al., 2017). The research question addressed here is how different bounds change the structure of the latent space, to provide better or worse disentanglement scores. Finally, we would also like to test our novel bounds on all the major benchmark datasets used for assessing the performance of VAEs and compare them with other state-of-the-art bounds on the model evidence.

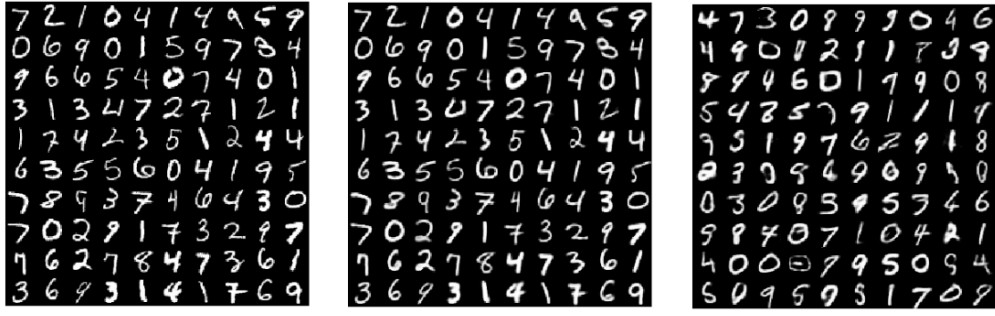

Figure 4: Method: IWAE with K=50 samples. From left to right: original binary MNIST test images, reconstructed and randomly generated ones.

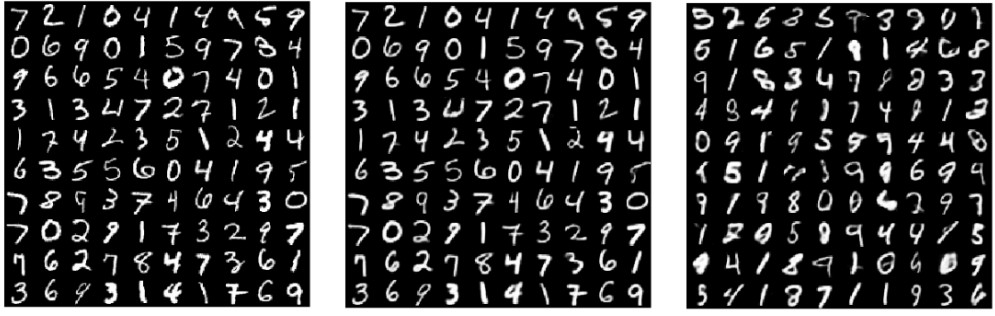

Figure 5: Method: qIWAE($\tau = 0.5$) with K=50 samples. From left to right: original binary MNIST test images, reconstructed and randomly generated ones.

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
