# OpenReview forum: "Variational autoencoders trained with q-deformed lower bounds"
_ICLR.cc/2019/Workshop/DeepGenStruct — DeepGenStruct 2019_

### Official Review · AnonReviewer2 · 2019-04-14
**Rigor and clarity lacking, but a reasonable contribution**

**Rating:** 3
**Confidence:** 3

**Review:**

This paper proposes a new lower bound for variational inference based on q-deformed logarithms, a generalization of the logarithm that augments it with a q parameter that controls its concavity. The authors train VAEs with this bound and report performance improvements over the ELBO, but not the IWAE bound.

The paper has issues with clarity, and rigor, but is overall a reasonable contribution.

Specific Feedback:

1. The derivation of the q-deformed lower bounds is lacking in rigor. The new bounds are just stated by swapping the q-logarithm for the standard logarithm without discussing whether that is possible. A proof that the new bound is a valid lower bound would be useful.

2. Similarly, it is not clear if swapping in the q-logarithm gives a lower bound on the log likelihood of the data (q=1) or the q-deformed log likelihood of the data for a specific value of q. The latter seems more likely. If that is the case, a discussion of the benefits and drawbacks of optimizing a lower bound on the q-deformed log likelihood of the data would be helpful to the reader.

3. In the actual training procedure the authors optimize bounds with different values of q for each batch. An argument should be made that this procedure is still optimizing a valid lower bound on the (possibly q-deformed) log likelihood.

4. The optimization procedure for q is not clearly stated. It seems like q* is set to make the qELBO evaluated with q=q^* match qELBO* as closely as possible. So perhaps the optimization procedure attempts to minimize (qELBO* - qELBO(q=q*))^2 w.r.t. q*. This should be stated clearly, and the optimization procedure should be clearly motivated.

5. In evaluation, what method is used to estimate log \hat{p}_x?

---

### Official Review · AnonReviewer1 · 2019-04-14
**Missing a few elements**

**Rating:** 2
**Confidence:** 2

**Review:**

In the paper, the authors suggest modifying various VAE bounds (ELBO, IWAE) by replacing logarithm by q-logarithms. The paper is missing a few steps to be satisfying in my opinion - at its heart, it takes a lower bound to the true data evidence, and upper bounds it using a q-logarithm with q<1.0. While the new estimate would be 'tighter' it if it was still a lower bound, the authors provide no guarantee that the resulting qELBO are still in fact lower bounds; maximizing them may therefore not make sense.
The q values used are very close to 1, which suggests the q-logarithm is used a mild hyperparameter to smooth the objective function; gains on experimental data are minor.

---

### Decision · Program_Chairs · 2019-04-19
**Acceptance Decision**

**Decision:**

Accept

**Comment:**

This paper has interesting contributions. However both reviewers agree it would be valuable to add an analysis as to whether the objective is a bound on the log marginal or not.